# An Overview of the Most Significant Zoonotic Viral Pathogens Transmitted from Animal to Human in Saudi Arabia

**DOI:** 10.3390/pathogens8010025

**Published:** 2019-02-22

**Authors:** Omar A. Al-Tayib

**Affiliations:** 1Abdullah Bagshan for Dental and Oral Rehabilitation (DOR), Dental College Research Center, King Saud University, Riyadh 12372, Saudi Arabia; altayibomar@gmail.com; Tel.: +966-543-430-285; 2Department of Pharmacology and Toxicology, Faculty of Veterinary Medicine, University of Khartoum, Khartoum 11111, Sudan

**Keywords:** zoonosis, viral pathogens, Saudi Arabia, MERS-CoV, AHFV, Hajj and Umrah, animal, public health

## Abstract

Currently, there has been an increasing socioeconomic impact of zoonotic pathogens transmitted from animals to humans worldwide. Recently, in the Arabian Peninsula, including in Saudi Arabia, epidemiological data indicated an actual increase in the number of emerging and/or reemerging cases of several viral zoonotic diseases. Data presented in this review are very relevant because Saudi Arabia is considered the largest country in the Peninsula. We believe that zoonotic pathogens in Saudi Arabia remain an important public health problem; however, more than 10 million Muslim pilgrims from around 184 Islamic countries arrive yearly at Makkah for the Hajj season and/or for the Umrah. Therefore, for health reasons, several countries recommend vaccinations for various zoonotic diseases among preventive protocols that should be complied with before traveling to Saudi Arabia. However, there is a shortage of epidemiological data focusing on the emerging and reemerging of zoonotic pathogens transmitted from animal to humans in different densely populated cities and/or localities in Saudi Arabia. Therefore, further efforts might be needed to control the increasing impacts of zoonotic viral disease. Also, there is a need for a high collaboration to enhance the detection and determination of the prevalence, diagnosis, control, and prevention as well as intervention and reduction in outbreaks of these diseases in Saudi Arabia, particularly those from other countries. Persons in the health field including physicians and veterinarians, pet owners, pet store owners, exporters, border guards, and people involved in businesses related to animal products have adopted various preventive strategies. Some of these measures might pave the way to highly successful prevention and control results on the different transmission routes of these viral zoonotic diseases from or to Saudi Arabia. Moreover, the prevention of these viral pathogens depends on socioeconomic impacts, available data, improved diagnosis, and highly effective therapeutics or prophylaxis.

## 1. Introduction

Rudolf Virchow (1821–1902), one of the foremost 19th century German leaders in medicine and pathology [1], noted a relationship between human diseases and animals and then introduced the term “zoonosis” (plural: zoonoses) in 1880 [2]. Later, the World Health Organization (WHO) in 1959 specified that “zoonoses are those diseases and infections which are naturally transmitted between vertebrate animals and man” [3]. Venkatesan and co-authors reported that the term zoonosis is derived from the Greek word “zoon” = animal and “noso” = disease [4]. Zoonotic pathogens causing different kinds of diseases are of major public health issues worldwide [5]. These zoonotic diseases include various infections such as viral, bacterial, fungal, protozoan and parasitic diseases shared in nature by man and animals (domestic and wildlife) [6]. Of these, an epidemiological study confirmed that about 61% of the total number of microbial diseases affecting man is zoonotic [7]. Moreover, another study suggested that animals are the major sources of human zoonotic infections, globally and that among all the emerging infectious diseases, almost 75% are considered to be caused by animals [8]. Thus, almost every year since the last two decades, a new virus has been emerging [9]. During the last three decades, rats have been increasingly implicated in several emerging and reemerging human outbreaks of zoonotic diseases and have accounted for ~75% of the new zoonotic diseases in nature according to several studies [10,11,12]. This constitutes about 61% of all communicable diseases causing illnesses in man [7,12]. Furthermore, a study suggested that some zoonotic diseases could affect the socioeconomic output globally. A report on the impact of foodborne zoonotic diseases estimated its costs at about $1.3 billion every year worldwide [13]. 

Since zoonotic diseases can easily be transmitted to man in several ways, they target persons who work closely with animals; this plays a big role in zoonotic transmission. Such persons working with animals include veterinarians, slaughterers/butchers, farmers, researchers, pet owners (e.g., through bites and/or scratches of owners of indoor pet-animals), and animal feeders in animal companies using animal products, via animals used for food (e.g., meat, dairy, eggs, birds, infected domestic poultry, and other birds). Furthermore, transmission can also occur through animal vectors (e.g., tick bite, and insects like mosquitoes or flea) [4,14,15]. In addition, in the transmission of bacteria in comparison with viruses, the role of contaminated food and water, the importance of international travels as well as changes in land use and agriculture, are important [4,9]. According to recent WHO data, more than 75% of the different zoonotic diseases that may cause illnesses in humans are transmitted through animals and/or animal products [16]. Nevertheless, several zoonotic pathogens may be transmitted from various animals to man via several direct and/or indirect pathways such as close contact with the infected animals that might be shedding the infectious pathogen, when humans use contaminated sources of food or water, and/or by outdoor or indoor animal scratches or bites [17,18,19,20,21]. The prevention and control strategies against zoonotic pathogens are considered important issues and a global challenge requiring efforts of all veterinarian and medical staff [22].

Animals have been domesticated for a long time in the Arabian Peninsula; and in Saudi Arabia, humans are living in close contact with animals [23]. The spread of any zoonotic disease in Saudi Arabia is considered to be of a high public health importance because it might put the Saudi Arabia peoples, as well as millions of Muslim pilgrims from the about 184 Islamic countries worldwide, at great risk. Moreover, thousands of animals of unknown origin are sacrificed daily during the annual pilgrimage for all pilgrims in Makkah [24,25,26,27]. Furthermore, pilgrims slaughter over one million sheep, cows, and camels in Mina to mark the successful completion of the Hajj [28], aside the 42,000 beasts that are slaughtered in Makkah abattoirs. While some people are at slaughterhouses to offer their sacrifices personally, others from nearby cities of Makkah, Jeddah, and Taif come to collect sacrificial meat [28,29].

Recently in Saudi Arabia, a huge number of new pet clinics and/or pet stores opened, selling all kinds of pets. They breed, shower, and clean pets, which bring major feeling of psychological well-being to the modern urbanized lifestyle of their Saudi Arabian owners. However, all these pet markets may need to be targeted by the Ministry of Health (MOH) and/or the Ministry of Agriculture because all kinds of pets—including cats, dogs, rodents, and monkeys—which are sold, may transmit several zoonotic diseases. In this review, we identified the most important and prevalent emerging and reemerging viral zoonotic pathogens in Saudi Arabia, taking into account the current incidence and prevalence of zoonotic diseases, the health situations, the zoonotic sources of human infection, and the current available control strategies that could prevent such infectious zoonotic diseases. In addition, we identified the primary sources of information on zoonotic pathogens in Saudi Arabia [30,31,32]. Data sharing and dissemination of significant findings could make a remarkable difference in the global control; it could provide useful information, particularly to Muslims on pilgrimage, when they travel to Saudi Arabia during Umrah seasons and/or the annual Hajj pilgrimage.

## 2. Viral Zoonoses

Frequent mixing of different animal species in the markets in densely populated areas, and the human intrusions into the natural habitats of animals, have facilitated the emergence of novel viruses. The most important zoonotic viral diseases of which eight were diagnosed (in dead or diseased animals or through antibody detection) on the Arabian Peninsula over the last years include rabies, Middle East Respiratory Syndrome (MERS-CoV), influenza virus (IFV), Alkhurma hemorrhagic fever, Crimean-Congo hemorrhagic fever (CCHF), Rift Valley fever (RVF), West Nile fever (WNV), and dengue fever virus. Among these eight zoonotic viral diseases, two (Alkhurma and MERS-CoV) were first reported in a patient in 1994 and 2012, respectively in Saudi Arabia [33,34]. These two were transmitted later to several other countries, not only in the Middle East but also to Africa, Asia, and Europe.

### 2.1. Rabies

Rabies is an almost invariably fatal zoonotic disease, which belongs to the genus Lyssavirus of the RNA family Rhabdoviridae. Rabies virus is considered an endemic viral infectious disease in animals in Saudi Arabia. Recent scientific data on rabies cases reported in camels at Al-Qassim region (one of the thirteen administrative regions of Saudi Arabia) showed that there is an increasing number of this fatal virus disease [35]. However, the most significant animal bites which have been recorded in Saudi Arabia were caused by different species of animals including dogs, cats, rodents, and foxes [36]. Later, Al-Dubaib reported rabies in dromedaries in Saudi Arabia in 2007 and suggested an incidence of about 0.2% for rabies that was reported among 48 camel herdsmen looking after more than 4000 animals [35].

Interestingly, another survey was conducted between 1997 and 2006 in the Al-Qassim region of central Saudi Arabia among 4124 camels and showed that about 0.2% of clinical rabies incidence is caused by dogs (may be cause it highly used as a perfect guard for camels), followed by foxes; furthermore, the diagnosis of viral rabies in that region was confirmed among 26 dogs, 10 foxes, 8 camels, and 7 cats [35]. Lately, the relevant government authorities (the MOH and Ministry of Agriculture in Saudi Arabia) in an updated report between 2007 and 2009 showed that there were a total of 11,069 animal bites to humans in Saudi Arabia [36]. Furthermore, most cases of animal bites were caused by dogs (49.5%) and cats (26.6%), followed by mice and rats, camels, foxes, monkeys, and wolves [36]. Moreover, dogs, particularly feral dogs and foxes, are considered the most important host for rabies virus; however, bats are also considered as reservoirs of this disease. Humans can become rabid by direct contact with animal mucosal surfaces via bites.

According to the MOH and Ministry of Agriculture data in Saudi Arabia, pets are responsible for most animal bites in humans [36], and it is well-known that insufficient vaccination coverage of pets are among the most common hallmarks of the endemic status of rabies worldwide [37]. More recently, many Saudi and expatriate families are keeping pets; however, there are limited number of specialized veterinary clinics (~5) within the Kingdom of Saudi Arabia that have fully licensed veterinary laboratories with state of the art technologies and veterinary staff.

Globally, almost 95% of all human deaths caused by rabies occur in Africa and Asia [38]. However, Saudi Arabia, as one of the Asian countries, has scarce publications and epidemic data on rabies status [38,39]. Moreover, Memish et al., between 2005 and 2010 in Saudi Arabia, reported the histologic detection of the virus by identifying Negri bodies in the brain samples of 40 animal rabies cases. The study showed that among the 40 suspected rabies cases, 37 (~92.5% of all cases) were found to be positive; thus confirming rabies cases among 11 dogs, 6 foxes, 6 sheep, 5 camels, 4 goats, 3 wolves, and 2 cows [36].

Furthermore, more recent data confirmed the transmission of rabies virus in Saudi Arabia by feral dogs [23]. In spite of these facts, there are very few studies available, and no case of human rabies has been reported in recent decades from Saudi Arabia [40]. However, in March 2018, a scientific work was reported as the first confirmed case of human rabies in Saudi Arabia from Makkah City, which has now been published [41]. Indeed, several previous global epidemiological data confirmed that rabies accounted for 24,000 to 60,000 human deaths per year [42], and more than 40% of these cases occur in children < 15 years of age [43].

In September 2016, a 60-year-old Saudi man, presented with different clinical features—such as nausea, vomiting, and epigastric pain, with significant features suggestive of gastritis—at Makkah hospital. His past medical history was significant for hypertension and diabetes type 2. During the clinical diagnostic procedure of this case, he developed respiratory distress and tachycardia, for which he was transferred to the intensive care unit [41]. Because, his case worsened with chest pain and ventricular tachycardia he was referred to the King Abdullah Medical City in Makkah for further management. The written diagnostic report indicated that he had acute anteroseptal myocardial infarction, had coronary angiogram which suggested that two-vessels were diseased with left main involvement, and surgical intervention was planned. After the decision for surgery, he was found to have leukocytosis and severe retching while attempting to drink water (hydrophobic behavior), which necessitated further review by the infectious disease consultants based on the patient’s clinical symptoms. The consultant team discovered the history of an unprovoked scratch on the patient’s face by a dog in Morocco a month prior to the admission at the hospital. Also, the patient stated that he only received tetanus vaccine. All diagnostic tests including neurologic examination were unremarkable and his saliva polymerase chain reaction (PCR) test confirmed rabies virus. He was administered Verorab rabies vaccine and human hyperimmune rabies immunoglobulin (20 IU/kg) intramuscularly (IM) [41]. In addition, he had troponin I (4.65 ng/mL), creatine kinase isoenzyme MB (CKMB) was found (30.08 ng/mL), and serum glucose (200 mg/dL). On the fifth day of hospital, he had recurrent episodes of ventricular tachycardia, progressively worsening of hemodynamic parameters, and he succumbed to his infection on that day. There is no vaccine against rabies recommended for travelers from/to Saudi Arabia, and no rabies treatment is offered to pet dogs. However, vaccination is given to dogs before they are infected; otherwise they are euthanized if infected.

According to a previous study, most patient injuries from animal bites in Saudi Arabia showed some variations due to the monthly incidence and/or, according to the animal species [36]. Bites by dogs and cats were reported frequently throughout the year, with a decrease in April and between August and October. However, bites by foxes increase between August and September while camel bites were more frequent between December and March of the subsequent year. The same previous study suggest that these seasonal variations of injuries might be due to the Saudi population habits, with people going to the desert for leisure activities during good weather periods. Laboratory diagnosis of rabies viral disease occur with the use of the rabies virus direct fluorescent antibody test (DFAT) on brain samples and hippocampal tissue [44].

While rabies is considered nearly 100% fatal, it is also 100% preventable, and thus vaccination to pets is the key element to prevent the risk of rabies zoonotic infection [45]. Reports of the epidemiology of rabies virus worldwide, and particularly in Saudi Arabia, suggest that it is on the increase, thus the implication of this virus’ potential to spread across borders from high to low prevalence countries was highlighted [23].

### 2.2. Middle East Respiratory Syndrome

The MERS-CoV infection is considered to be a new respiratory disease with a dire global concern [46]. MERS-CoV infections are caused by a newly emerging coronavirus (CoV), belonging to the designated lineage C of *Betacoronavirus* of the RNA family Coronaviridae. With respect to viral origin and transmission, bats are thought to be the reservoir host of *Betacoronaviruses*, and the African *Neoromicia* bats in particular are the natural reservoir of MERS-CoV [47,48].

Since its emergence in 2012 in Saudi Arabia, when an elderly patient (60 years old) with respiratory illness died after admission to a hospital in Jeddah [34], the disease was subsequently reported to have been transmitted to several countries worldwide, and has affected more than 1000 patients with over 35% fatality [46,49,50,51].

Moreover, a 60-year-old Saudi man was admitted to a private hospital in Jeddah, Saudi Arabia in June 2012 with a history of fever, severe acute respiratory syndrome with cough, expectoration, and shortness of breath. He did not smoke; and for the disease, which was suggested to be due to an animal transmission of coronaviruses, he was treated with oseltamivir, levofloxacin, and piperacillin-tazobactam. On day 11, he died [34]. After this, a 61-year-old Saudi male with hypertension and diabetes with no history of smoking, reported for surgery. At the time of admission, he was asymptomatic. He was initially screened using nasopharyngeal swab, endotracheal aspirate, and serum sample for MERS-CoV per protocol with the MERS RRT-PCR assay. The results confirmed MERS-CoV infection. He died three days after admission. It was discovered that the patient owned a dromedary camel barn in Saudi Arabia, and had a history of close contact with camels, as well as a habit of raw milk consumption of an unknown duration [51].

Two studies have suggested a relationship between the infection and contact with dromedary camels [52,53]. In addition to this, serological diagnostic methods have been used to confirm MERS-CoV infections in dromedary camels for at least 2–3 decades and has thus confirmed camels as an intermediate host for this virus [54,55]. Thus, in 2012, a novel coronavirus (MERS-CoV) was isolated from two fatal human cases in Saudi Arabia and Qatar; and since then, more than 1400 clinical cases of MERS-CoV have been identified, and the great majority of the cases were from Saudi Arabia [56]. This previous report author raised a thoughtful comment related to the emerging viral diseases “Why We Need to Worry about Bats, Camels, and Airplanes” [56]. Moreover, another study suggested that MERS-CoV infection is usually transmitted from human’s direct contact with dromedary camels, especially when people drink the milk or use camel’s urine for medicinal purposes [57]. More recently, a metagenomics sequencing analysis of nasopharyngeal swab samples from 108 MERS-CoV-positive live dromedary camels marketed in Abu Dhabi, United Arab Emirates, showed at least two recently identified camel coronaviruses, which were detected in 92.6% of the camels in that study [58]. However, limited human-to-human infections have been reported.

The prevalence of MERS-CoV infections worldwide still remains unclear. In addition to this, the WHO reported about 1797 cases of these infections since June 2012, with about 687 deaths in 27 different countries, worldwide. Recently, a study was conducted from June 2012 to July 2016, during which samples were collected from MERS-CoV infected individuals, from the National Guard Hospital in Riyadh (the Saudi Arabian capital city), the MOH in Saudi Arabia, and other Gulf Corporation Council countries, to determine the prevalence of MERS-CoV [59]. The epidemiologic data that were collected, showed that the highest number of cases (about 1441 of 1797 patients) were reported from Saudi Arabia (~93%). Among the 1441 MERS-CoV cases from Saudi Arabia, Riyadh was the worst-hit area with 756 infected cases (52.4%), followed by the western region of Makkah where 298 cases (20.6%) were reported [59]. 

Furthermore, this study also showed that the incidence of MERS-CoV infections was highest among elderly people aged ≥60 years [34,59]; with speculation that there might be certain conditions or factors involved. It is considered that MERS-CoV infection might have a peculiar gender predisposition [60]. Recent data examined the mortality in patients with MERS-CoV and the gender relationships, looking at the survival of cases among females and males. It was suggested that males have a higher risk of death [61,62]; however, this was contradicted by the findings from two other studies which suggested that males have a low risk of death [63]; while another survey which examined the influence of gender on 3-day and 30-day survival, found a low risk of death especially in the older age group [64]. On the other hand, Badawi et al., suggested that MERS-CoV infections could be mild and may only result in death among patients suffering from any kind of immune system disorder and/or any chronic disease [46].

More recently, data regarding the mortality in patients with MERS-CoV have been published. According to Saudi Arabia’s MOH daily statements, dated from February 26 through March 3, laboratory-confirmed new cases of MERS-CoV and 2 deaths occurred [65]. Recently, on February 26, patients infected while hospitalized at Riyadh included two men (23 and 59 years old) in stable condition, who were not healthcare workers. According to a February 27 update, a new case involved a 71-year-old man from the city of Buraydah who later died. Meanwhile, on March 1, another MERS-CoV infection in a Riyadh hospital patient, a 64-year-old man who was listed in critical condition and who likewise had contact with camels, as the other two patients, was reported. Thus, the MOH stated that the spillover from camels is thought to be the main source of MERS-CoV in Saudi Arabia, since all these patients were exposed to the animals before reporting ill [65]. 

Furthermore, an 83-year-old patient from Riyadh, and other two patients who had camel contacts from Hail city in the north central part of Saudi Arabia were listed in critical condition. The illness in these patients was reported on March 1. According to a March 3 statement, another patient, a 74-year-old man from Najran located in southern Saudi Arabia, was reported. The man was listed in a stable condition. Of these new cases, only one death, involving the 83-year-old man from Riyadh, according to the March 3 MOH statement, was reported [65]. Still, much work is needed to detect the MERS-CoV infection risk in Saudi Arabia, because data showed increasing number of cases exist among the eight countries including Saudi Arabia. Thus, the emergence of MERS-CoV in the region and its continuing transmission from 2012–2017, currently poses one of the biggest threats to global health security [66]. Most cases (over 85%) reported to date have been from countries in the region (e.g., Egypt) notably from Saudi Arabia, with 1527 cases including 624 deaths [67].

### 2.3. Influenza

Influenza viruses are considered to be important infectious viral diseases, which is caused by three virus types (A, B, and C) [68]. Due to their zoonotic spread, influenza type A infects both humans and animals, and causes moderate to severe illness, with more likelihood of fatalities in young children and the elderly [69,70]. Other types of influenza, including type B and C, infect only humans [71]. Furthermore, influenza A viruses, members of the RNA family Orthomyxoviridae, are further classified into human, swine, and avian influenza viruses. However, during the 1918 influenza pandemic, swine influenza virus infected one-third of the world’s population (an estimated 500 million people) and caused approximately 50 million deaths [72]. 

Since 2006, several infections with this virus have been recorded from various areas worldwide, including Saudi Arabia [73]. At the end of April 2009, an outbreak of a new type of influenza, A/H1N1, started in Mexico and the USA [69]. The WHO declared the pandemic influenza A (H1N1) as a “public health emergency of international concern” following the first few initial cases in Mexico, and subsequently in the USA [69,74]. In Saudi Arabia, the epidemiological data for influenza virus were collected using a predesigned questionnaire with the first 114 confirmed pandemics influenza A (H1N1) cases identified by the Infectious Diseases Department from the MOH, and the database during the period covered from 1 June to 3 July 2009 [72]. However, according to the Saudi MOH data, the number of laboratory-confirmed cases of the virus in Saudi Arabia as at 30 December 2009 was 15,850, with 124 deaths [72]. The virus later spread worldwide, causing a pandemic, and the most recorded cases then, as reported by the WHO in the Middle East, were in Saudi Arabia with 14,500 cases, followed by Kuwait [69], Egypt, and Oman; with less number of infected patients [68,75]. Nevertheless, between 1979 and 1980, a serosurveillance outcome of swine influenza virus from Egypt provided evidence of laboratory diagnosis and very early confirmation of the virus in human patients [76]. In Saudi Arabia, the influenza surveillance system has been established since 2004. Moreover, among people with certain chronic medical diseases or conditions, a trivalent influenza vaccine (TIV), which contains inactivated antigens for two different subtypes of influenza viruses (types A and B), became available in Saudi Arabia [72].

Indeed, H1N1 is now in the post-pandemic period and has become a seasonal influenza virus that continues to circulate with localized outbreaks of varying magnitude in Saudi Arabia [77]. A previous data was collected using a predesigned questionnaire for the first 100 cases of pandemic influenza A (H1N1) from different hospitals in Saudi Arabia. The age of patients enlisted in the data ranged from 1 to 56 years. The age groups with the highest percentage of cases were between: 20 and 30 years (35%), and 1 and 10 years (22%). There were 45 males and 55 females, and 53% patients had some contacts with infected persons within Saudi Arabia while about 47% had history of travels into Saudi Arabia and/or the Philippines [72]. These facts are similar to the previous relationship noted between the occurrence of zoonotic viral diseases and the gender of patients and/or their ages, as reported for another viral (e.g., MERS-CoV) infection; provided certain conditions are met [60,61,62,63]. Interestingly, among elderly patients, influenza cases were higher in females than males. This relationship with viral infection occurring particularly with respiratory viral diseases, might pave the way and play a big role of more significant importance in the detection of these diseases, taking into account the influence of climate change and the different environmental factors [69,77,78].

Nevertheless, between September 2013 and October 2014, about 406 samples were taken from several patients presenting with respiratory symptoms to King Abdulaziz University Hospital, Jeddah, Saudi Arabia. However, during this study conducted to detect the susceptibility to the influenza viruses circulating in the western part of Saudi Arabia, out of all the tested samples, 25 (6.2%) respiratory samples were positive for influenza H1N1 virus, 1 (0.25%) was positive for influenza H3N2 virus, and 7 (1.7%) were positive for influenza B virus [78]. Furthermore, H1N1, now in the post-pandemic period, has become a seasonal influenza virus that continues to circulate with localized outbreaks of varying magnitude [71]. Interestingly, the presentation of influenza virus infections in humans usually vary from mild, self-limiting respiratory-like illness, to severe cases that may result in death [70,79]. Nevertheless, a recent study has shown that subclinical infection in human exists, as revealed by the serological surveillance [76,80]. Therefore, the epidemiological surveillance of influenza in Saudi Arabia is highly important especially with the fact that influenza cases have also been highly reported can spread globally [78]. Thus, geographic influences on influenza virus infection in Saudi Arabia must be of concern [78]. This is relevant because a remarkably high number of Egyptian Muslims visit Saudi Arabia yearly to participate in the Umrah and/or the Hajj pilgrimage; in addition, the impact of the poultry industry in Egypt is also worth considering, with an estimated 1 billion birds and several millions of engaged laborers with/without surveillance [81].

It is well-known that the influenza drugs, antiviral agents, and the current seasonal influenza vaccines are effective in reducing the incidence and severity of the disease, sickness, and/or complications. However, the important strategy for influenza management includes the provision of prophylaxis and treatment [78]. However, it is possible for widespread drug resistance against antiviral agents or vaccines to emerge in patients who extensively abused the drugs, in addition to those who have never received such treatment, globally [82,83,84]. Furthermore, influenza viruses pose a challenge to vaccine developers and manufacturers due to the fact that these viruses are continually changing in nature, including hemagglutinin and neuraminidase [78,85]. Moreover, while resistance to neuraminidase inhibitors (e.g., oseltamivir and zanamivir) have been reported to sporadically occur, the resistance to oseltamivir has been widely reported since 2007, with a worldwide spread [86]. This highlights why there is an urgent need for the public health system to monitor continuously via globally active influenza surveillance programs. Furthermore, there is need to monitor the circulating influenza viruses strains, as well as the occurrence of any resistance, using appropriate diagnostic methods. This is considered highly essential in Saudi Arabia. Interestingly, survey data has shown an increasing report of the viral infection from Egypt, since Hajj Egyptians has ranked in the top 10 list of countries with the highest number of Mecca pilgrims in the last 10 years [81]. Influenza is highly susceptible to antiviral drugs such as oseltamivir, according to a more recent epidemiological study [87]. Although millions of Muslims, globally, travel annually to Saudi Arabia to perform Hajj and/or Umrah in the holy places including both Makkah and Al-Madinah for very limited period (~10 days), this gathering could play a major role in the introduction of new influenza viruses, not only to Saudi Arabia but also to the rest of the world [83,87]. Unfortunately, there is no such influenza surveillance program in Saudi Arabia, thus this pose a serious public health concern. 

Recently, in a study of 1600 pilgrims screened on arrival at the 2010 Hajj season in Saudi Arabia, 120 (7.5%) had influenza A virus (9 out of the 120 had H1N1 virus) [88]. Additionally, the epidemiological data showed that the pilgrims had the potential not only to introduce these viruses to Saudi Arabia, but also to export the influenza virus back to their home countries [78,89]. This can occur in Saudi Arabia, despite the availability of a TIV containing inactivated antigens for influenza virus types A and B, which can protect against the influenza virus infection [72].

Importation of resistant and highly pathogenic viruses including influenza viruses can occur worldwide. Despite this, there is lack of studies and data on drug susceptibility, and a very limited number of studies and reports on viral isolates, except for one study conducted in Jeddah, according to the best of our knowledge [78]. Most importantly, this highlights the importance of circulating influenza viruses in Saudi Arabia, hence there is need to ensure effective use of antivirals for prophylaxis and treatment of influenza. Furthermore, the rate of vaccination against influenza is very low among pilgrims and healthcare workers [78,90]. Moreover, studies are needed to provide a clear picture on the impact of drug resistance on Saudi Arabia’s endemic pathogens, including the influenza viruses. In a recent communique, the Ministry of Environment, Water, and Agriculture for Saudi Arabia reported two cases of H5N8 avian influenza in the Kharj Governorate [91]. The latest update by the Ministry revealed that the number of samples collected from Saudi regions since the start of the influenza outbreak had reached 12,829. Positive results from samples and laboratory tests indicate 171 positive cases, and the Saudi authorities have taken action by culling as many as 254,050 birds within a 24-h period [91]. 

In contrast, several epidemic zoonotic cases of influenza H5N1 have been reported in domestic cats in several countries in Asia, Europe, the USA, and Italy [87]. Moreover, the epidemic of influenza in dogs might be related to a serious public health issue and could be shown to have resulted from zoonotic diseases from pets, similar to the avian influenza H3N2 outbreak reported in pet dogs in South Korea in 2007 [92]. Nevertheless, a recent study has shown that the role of pets, particularly cats and dogs in the epidemic of influenza as a source of human infection seems limited. However, cats were shown to be fully susceptible to experimental infection, and infected cats were able to infect naive cats [87]. In 2009, pandemic H1N1 infection in a domestic cat in the USA from Iowa was diagnosed by a novel PCR assay; thus, human-to-cat transmission was presumed [93]. Despite this prior evidence, the role of pets including cats and dogs seem even more limited in the dispersal of avian influenza to humans. Rather, humans may be the source of pet infection, as suggested for influenza H1N1 and/or H3N2 virus infections [87,92,93,94].

Most importantly, epidemic zoonotic cases of influenza among pets has highlighted the importance of circulating influenza viruses globally; especially, to ensure the effective use of antivirals for the prophylaxis and treatment of influenza, in particular, with the increase in the number of pets stores in Saudi Arabia, especially in Riyadh [78,90]. Surprisingly, previous data focused on the occurrence of zoonotic infection of different influenza virus types, and particularly, the transmission of avian influenza virus H3N2 to domestic dogs [92]. Several studies have examined and confirmed the occurrence of zoonotic infection of the influenza A virus H1N1 pandemic, especially in domestic cats [93,94]. Nevertheless, epidemiological studies on different zoonotic infections among the pets in Saudi Arabia including cats, dogs, and/or baboons are very rare. However, a previous case report confirmed a relationship between some zoonotic diseases causing respiratory symptoms, such as influenza, among pets [95]. This study suggests that severe lung infection with dry cough and severe anemia should lead to the suspicion of a secondary infection with zoonotic balantidiasis, which infected a hamadryas baboon from Saudi Arabia in a research center for pets in Riyadh [95]. Furthermore, two other epidemiological zoonotic study on *Balantidium coli* protozoan zoonotic infection in camel was reported from Riyadh [31]. 

In addition, another previous data confirmed the occurrence of *Toxocara canis* zoonotic infection based on respiratory symptoms reported at the pet clinics in Saudi Arabia (and also in Riyadh where the symptoms occurred in dogs) [30]. Still, more such studies are needed to highlight the important issues and/or provide clearer pictures of the zoonotic pathogens among pets in Saudi Arabia; however, pet ownership has been growing rapidly as well as the number of pet stores among the Saudi population.

### 2.4. Alkhurma Hemorrhagic Fever Virus

Alkhurma hemorrhagic fever virus (AHFV) in humans was discovered in 1994 [33]. The first case reported in a butcher from the city of Alkhurma, a district south of Jeddah in Saudi Arabia, died of hemorrhagic fever after slaughtering a sheep. The viral infection has a reported fatality rate of up to 25% [96]. Interestingly, one of the previous reports regarding this disease showed a misunderstanding of the real name of this infection, called Alkhurma, not Alkhumra [97,98]. Because subsequent cases were diagnosed in patients from the small town known as Alkhurma in Jeddah from where the virus got its scientific name; the name was accepted by the International Committee on Taxonomy of Viruses [99]. Thus, based on evidence, the first case was confirmed to be the butcher, following the slaughtered sheep [100]. Therefore, a study was conducted among affected patients to address this disease as a public health issue. Blood samples were collected from household contacts of patients with laboratory-confirmed virus for follow-up testing by enzyme-linked immunosorbent serologic assay (ELISA) for AHFV-specific immunoglobulin (Ig) G. Samples from persons seeking medical care were tested by ELISA for AHFV-specific IgM and IgG using AHFV antigen. Viral-specific sequence was performed by reverse transcription PCR (TiBMolbiol, LightMix kit; Roche Applied Science, Basel, Switzerland). A total of 11 cases were identified through persons seeking medical care, whose illnesses met the case definition for AHFV, and another 17 cases were identified through follow-up testing of household contacts [100].

Subsequently, the virus was isolated from six other butchers of different ages (between 24 and 39 years) from the city of Jeddah, with two deaths. The diagnosis was established from their blood sample tests. The serological tests later confirmed four other patients with the disease [101]. From 2001 to 2003, the study on the virus initial identification in the city of Alkhurma again identified 37 other suspected cases; with laboratory confirmation of the disease in 20 (~55%) of them. Among the 20, 11 (55%) had hemorrhagic manifestations and 5 (25%) died [102]. The virus was later identified in three other locations: from the Western Province of Saudi Arabia (*Ornithodoros savignyi* and *Hyalomma dromedarii* were found by reverse transcription in ticks) and from samples collected from camels in Najran [103,104]. AHFV virus was considered as one of the zoonotic diseases; however, the mode of transmission is not yet clear. Recently, it was suggested that the disease reservoir hosts may include both camels and sheep. The virus might also be transmitted as a result of skin wounds contaminated with the blood or body fluids of an infected sheep; through the bite of an infected tick, and through drinking of unpasteurized or contaminated milk from camels [101,105]. 

In humans, this zoonotic disease may present with clinical features ranging from subclinical or asymptomatic features to severe complications. It is related to Kyasanur Forest disease virus, which is localized in Karnataka, India [106,107]. However, epidemiologic findings suggest another wider geographic location for the disease in western (including Jeddah and Makkah) and southern (Najran) parts of Saudi Arabia, and the virus infections mostly occur in humans [96,101,102]. A study was conducted by Alzahrani et al. in the southern part of Saudi Arabia particularly in the city of Najran (with populations of ~250,000), an agricultural city in Saudi Arabia, where domestic animals are reared at the backyard of owners. After the initial virus identification, from January 2006 through April 2009, 28 persons with positive serologic test results were identified. Infections were suspected if a patient had an acute febrile illness for at least two days; when all other causes of fever have been ruled out [101]. Additionally, data analysis indicated that patients infected with the virus were either in contact with their domestic animals, involved in slaughtering of the animals, handling of meat products, drinking of unpasteurized milk, and/or were bitten by ticks or mosquitoes. Symptoms consistent with AHFV infection—including fever, bleeding, rash, urine, color change of the feces, gum bleeding, or neurologic signs—then develop [95]. Fortunately, infected patients responded to supportive care (including intravenous fluid administration and antimicrobial drugs when indicated), with no fatal cases.

In summary, AHFV is a zoonotic disease with clinical features ranging from subclinical or asymptomatic features to severe complications. Another study highlighted different characteristics of the exposure to the blood or tissue of infected animals in the transmission of AHFV to humans. Of the 233 patients confirmed with infections, 42% were butchers, shepherds, and abattoir workers, or were involved in the livestock industry [108]. More recently, a study on infection using C57BL/6J mice cells showed that the clinical symptoms of the disease were similar to the presentations in humans [109]. However, Alkhurma disease resulted in meningoencephalitis and death in Wistar rats, when high titers to the infection occurred [98]. In addition, exposures to mosquito bites are regarded as potential sources of transmissions of the infection; however, very few available data support this [97]. Although, available data shows that Alkhurma virus has been isolated following mosquito bites [102]. However, another study suggested that mosquitoes may play a role only as a vector in the transmission of the disease [100].

### 2.5. CCHF

CCHF is a zoonotic viral disease from the Bunyaviridae family, and the principal vector for the disease is ticks of the genus *Hyalomma.* It is most commonly endemic in Africa, Middle East, Asia, and Eastern Europe [110,111]. It is an acute, highly-contagious, and life-threatening vector-borne disease responsible for severe hemorrhagic fever during outbreaks, and a fatality rate of up to 40% [112,113]. The infectious disease was recognized first in the Crimean Peninsula in 1944, and it was named Crimean hemorrhagic fever virus because the virus was isolated for the first time from a febrile child in 1956 from Stanleyville (now Kisangani), Democratic Republic of Congo [114].

Currently, the virus infects both humans and animals following tick bites [115]. However, a human can be infected by the animal through contact with the blood or tissues of the infected animal, in particular, exposures at the abattoirs are common. Therefore, workers in contact with animals (e.g., veterinarians, farmers’ and workers in slaughterhouses) form a high percentage of those affected [87]. Also, different species of infected animals—such as camels, cattle, sheep, goats, and ostriches—might be infected with no clinical signs [83]. In addition, human-to-human transmission is also documented, mostly through a form of nosocomial or in-house infection [113,116,117,118].

Lately, antibodies to the virus have been detected in different animal species, as reported in 1976, in Egyptians animals’ sera [119]. The preliminary seroepidemiological survey detected antibodies to the virus in 8.8% of camels’ sera and 23.1% of sheep sera, but no antibody was detected against the virus in the sera of other animals such as donkeys, horses and mules, pigs, cows, and buffaloes [119]. The epidemiology and distribution of CCHF in Saudi Arabia are unclear, but there are reports of CCHF as a result of the trading and importing of infected livestock from neighboring countries to Saudi Arabia [120]. 

In 1990, the CCHF virus (CCHFV) caused an outbreak involving seven individuals in Makkah, although the virus had not been reported previously in Saudi Arabia. Therefore, a study on the epidemiology of this virus was carried out in Makkah, Jeddah, and Taif from 1991–1993. About 10 out of 13 different species of ticks that were capable of transmitting the disease were collected from camels, cattle, sheep, and goats, but camels had the highest rate of tick infestation (97%), and *H*. *dromedarii* was the commonest tick (70%). An investigation in Makkah between 1989 and 1990, which included a serological survey of abattoir workers in contact with sheep blood or tissue, identified 40 human cases of confirmed or suspected CCHF with 12 fatalities [120]. The report from the investigation stated that the virus might have been introduced to Saudi Arabia through the Jeddah seaport via infected ticks on imported sheep; since then, it has been endemic in the Western Province of Saudi [120,121]. In addition, another previous study confirmed that the highest seropositivity rate of the virus in Saudi Arabia localities was associated with animals imported from Sudan [121].

Furthermore, the WHO reported 22 countries with CCHF including Saudi Arabia; however, all the remaining countries are either close to Saudi Arabia or are Islamic countries with high numbers of Muslims who travel annually to Saudi Arabia for Hajj pilgrimage. The same WHO epidemiological data suggest that in these 22 countries including Saudi Arabia, in recent years, there has been report of steadily increasing number of sporadic human cases, incidence, and outbreaks of the virus [122]. Furthermore, another study by WHO investigating CCHFV in the Eastern Mediterranean Region (EMR) stated that CCHF is a clear and growing health threat in the WHO EMR. Cases are being reported in new areas, showing a geographical extension of the disease that is probably linked to the livestock trade and the spread of infected ticks by migratory birds. According to ecological models, the increase in temperature and decreased rainfall in the WHO EMR could have resulted in the sharp increase in distribution of suitable habitats for *Hyalomma* ticks and the subsequent drive of CCHFV infection northwards [123].

Jazan Province, the Red Sea port city on Saudi Arabia’s southern border with Yemen, serves as the east–west portal from sub-Saharan Africa at Djibouti and the south–north route across the Yemeni frontier. It is a heavily traveled corridor for humans and animals entering Saudi Arabia, particularly during the annual Hajj pilgrimage. In November 2009, a total of 197 (19%) enrolled soldiers reported symptomatic illness during deployment, 49 (25%) of whom were hospitalized. Reported signs and symptoms included fever (n = 81), rash (n = 50), and musculoskeletal complaints (n = 128). A surveillance study was conducted to detect the causes of the several outbreaks through that area, which was reported as endemic over a wide geographic range. From the surveillance, serologic testing for CCHFV, AHFV, DENV, and RVF was completed for 1024 Saudi military units from several Saudi Arabian provinces. These units were previously stationed in other parts of the country, and were deployed to Jazan Province; the initial screening for IgG of each of these viruses was conducted by IgM testing for all IgG-reactive samples. Among the samples from all military forces, the study identified 40 reactive serum samples with a combined seroprevalence of 3.9 cases/100 soldiers tested. A confirmed serologic status of 1024 soldiers who were evaluated for IgG and IgM ELISA reactivity against CCHFV, RVF, AHFV, and DENV infections were positive for 6, 20, 13, and 1 sample, respectively [124].

### 2.6. RVF

RVF is a common arbovirus zoonotic disease caused by the RVF virus. The virus belongs to the genus *Phlebovirus* and family *Bunyaviridae*. It is most common in domestic animals, and causes mild to life-threatening infections in humans. The name of the disease was derived from the Great Rift Valley of Kenya, when the disease was described for the first time in 1912 [125]. Epidemiological tests have since been described after a highly fatal epizootic occurred there in 1930 [126].

RVF is a viral zoonosis with evidence of widespread occurrence in humans and animals in Africa and the Arabian Peninsula. The epidemiology of this virus in Saudi Arabia might be closely related to the ecological factors that are prevalent, as shown from another area, along the Great Rift Valley, which traverses Ethiopia and Kenya to northern Tanzania with the drainage ecosystems [127]. 

Saudi Arabia has many of the world’s mosquito vectors of parasitic and arboviral diseases. However, few studies have addressed their geographic distribution and larval habitat characteristics [128]. There are complex interactions between these factors that significantly impact mosquitoes ecological fitness and vectorial capacity for disease transmission, with important implications for vector management and control at the local and regional levels [129,130]. Therefore, studying these factors for different mosquito fauna will help in monitoring potential modifications of larval habitats due to rains, global climate change, or man-made activities. Previous studies on the ecology, distribution, and abundance of mosquito species in Kingdom of Saudi Arabia are generally few and sporadic; and most of these studies were conducted in the western and southern regions. These studies were conducted in the Asir Province in 1993–1995 and 1999–2001 [131,132]; Jeddah in 2007–2008 [133]; Makkah, Jeddah, and Al-Taif in 2004–2006 [134]; Madinah in 2004–2006 [135] and 2008–2009 [135,136]; then Najran Province in 2005–2006 [137]; Tehama Red Sea coastal plain in 2007–2008 [138]; and Makkah in 2009–2010 [139]. Fewer studies were conducted in the eastern region oases in 1979–1980 and 2009–2010 [140,141]; and in the central region (Riyadh) in 2002–2003 and 2003–2005 [142,143]. These studies reported the presence of many species from many genera, the most important of which are *Anopheles*, *Aedes*, and *Culex*. Among these studies, only a few provided the description of habitats of the larvae of these vectors. Even fewer studies provided evidence on the active role of some species on disease transmission; the existing ones were mainly for Anopheles vectors of malaria [138,144,145], as well as *Aedes* and *Culex* vectors of arboviruses such as Sindbis and dengue fever [141,146,147].

RVF is not considered a major type in the arboviruses family, which mostly are adapted to a narrow range of vectors; however, among this family, the RVF infection has a very wide range of vector including mosquitoes such as *Aedes* and *Culex*, flies, and often, ticks [148]. Interestingly, for different RVF species, RVF vectors have special roles about how they sustain the transmission of the disease ecologically to humans [149]. In some cases, the impact of rainfall, soil type, water, the persistence of breeding, and often wind, have significant effect on vector distribution [150]. Epizootics studies indicate that RVF disease follows unusually severe rainy seasons, a situation that may likely favor the breeding of a very large insect population, needed as a vector prerequisite.

Globally, RVF epidemiology was first reported in Africa with the 1989 RVF epizootics in Kenya when laboratory test reports confirmed virus isolation [151,152,153]. In 2000, the disease, for the first time, affected humans and livestock outside Africa, with the larger RVF disease incidence following outbreaks, reported in Saudi Arabia [154] and Yemen. Lately, RVF infections have been associated with minimal genetic diversity, epidemiologically; which has lately been considered to be a newly introduced single lineage of RVF viral disease [155]. Epidemiological reports from both Saudi Arabia and Yemen showed that the outbreak, which occurred in 2000, resulted in about 2171 human infections, and 245 deaths [156]. Furthermore, the fatality rate reported in southern Saudi Arabia then, reached 14%, and was considered the most severe epidemic in that area ever since [157]. Moreover, the disease outbreak was thought to have been transmitted in countries such as Saudi Arabia by infected imported ruminants from East Africa via the port of Djibouti and probably from Kenya and/or Sudan [121]. However, the fact remains that the RVF epidemic has been around for more than 70 years, with infections occurring at prolonged intervals in Eastern and Southern Africa [158,159]. Consistent with this, another report showed that the same virus strain was implicated in the 1997–1998 RVF outbreaks in Kenya and the 2000 outbreaks in Saudi Arabia and Yemen [130]. The outbreaks in Kenya later resulted in about 89,000 human infected with about 478 patients deaths [127,160].

Surprisingly, in 2000, Jup et al. found the mosquito species that was identified as a potential vector, which led to the assumption that the zoonotic viral disease in Saudi Arabia was transmitted by *Culex tritaeniorhynchus* [161]. Other species of mosquitoes were implicated in the transmission of this viral disease in other countries closer to Saudi Arabia [162,163,164]. Furthermore, another study reported the unexplained RVF virus infection among people from Saudi Arabia, with isolation and genetic virus characterization associated with illness in livestock, along the southwestern border of Saudi Arabia in September 2000 [164]. The study reported that vertical transmission of the virus in the epidemic mosquito vector occurred in Saudi Arabia. In addition, the study stated that the most abundant culicine mosquitoes collected were *Aedes vexans arabiensis*, *Culex pipiens complex*, and *Culex tritaeniorhynchus,* which were considered to be the most important epidemic and epizootic vectors of RVF virus in Saudi Arabia [164,165]. However, the same study, focusing on a very important issue which occurred during the rainy seasons; suggested that *Aedes vexans arabiensis* has the potential to be an important epidemic and epizootic vector because of the tremendous numbers of individual mosquitoes produced after a flood [164].

Characteristically, once the virus is introduced into permissive ecologies, it becomes zoonotic; thus, they are able to enhance vulnerability of the area to periodic outbreaks, with the potential to spread further into non-endemic environments with favorable conditions [166,167]. Saudi Arabia is considered a region where RVF virus has circulated actively. Noticeable data regarding zoonotic infection from animal to human from the Arabian Peninsula including Saudi Arabia has recently showed that it may be due to the consumption of unpasteurized camel milk [32,159,168]. Wernery reported *Camelus dromedarius* as the animal host and/or reservoir of RVF zoonotic infection, which was diagnosed in the Arabian Peninsula [23]. Due to the scientific data regarding RVF disease, it is quite clear that globalization of trade and altered weather patterns are a concern for the future spread of more infections, since the causative agent of this viral disease is capable of utilizing a wide range of vectors for its transmission. Thus, this poses a significant challenge to outbreak prediction, with inherently complex methods of infection control; therefore, mitigation and management of the virus will require concerted efforts [121,169,170].

### 2.7. Dengue Hemorrhagic Fever

Dengue hemorrhagic fever (DHF) viral disease is a serious global mosquito-borne infection. The clinical manifestation ranges from mild febrile illness to severe sickness which may include dengue shock syndrome [171]. The DHF virus belongs to the genus *Flavivirus* in the Flaviviridae family, which can usually be spread by mosquitoes of the genus *Aedes aegypti,* but less often through the genus *Aedes albopictus* [172,173]. Also, this virus is a single-stranded positive-sense RNA virus that exists as four different serotypes (DEN-1, DEN-2, DEN-3, and DEN-4) [174].

In Saudi Arabia, the disease is limited to the western and southwestern regions, such as Jeddah and Makkah where *Aedes aegypti* exists. However, all DHF cases in Saudi Arabia presented as a mild disease [171,175]. In fact, the first experience of DHF virus isolation from Saudi Arabia was recorded during an outbreak of the virus in 1994 [176], where the 289 confirmed cases reported in Jeddah were caused by DENV-2 [176,177,178]. However, during this first outbreak, in both summer and rainy season, at the end of the year, both DENV-2 and DENV-1 were isolated. In 1997, during the rainy season in Jeddah, there was an emergence of the DENV-3 virus [179]. In subsequent years, from 1997–2004; the emergence of DHF occurred with the three identified serotypes (DENV-1, DENV-2, and DENV-3) isolated in Jeddah [171]. Khan et al. reported the first outbreak of DHF viral infection in Makkah in 2004, and the isolated virus serotypes were DENV-2 and DENV-3 [179]. To prevent this outbreak, the Saudi Preventive Department in the MOH launched a comprehensive plan to control the disease [180], but other outbreaks occurred in Jeddah during the winter seasons of 2005 and 2006 [171,181]. However, Egger suggested that the reemergence of the disease in Saudi Arabia might be explained by the growing levels of urbanization, international trade, and travels [182].

In keeping with the findings of most previous studies, the epidemiological occurrence of DHF infection using the Saudi’s national data indicated that the majority (68%) of patients with dengue virus infection were Saudi nationals [183]. On the contrary, from the epidemiological report based on Saudi’s national data in previous publications, an estimated 15% of patients with DHF presented in Jeddah [184]. Kholedi et al. reported a higher percentage among patients with DHF infections in Jeddah, with about 51.9% and 48.1% among Saudi and non-Saudi patients, respectively. It was concluded that DHFV resurged sharply in Jeddah in 2004 with the number increasing dramatically to 1308 cases in 2006 [185]. Alzahrani et al., in another study, suggested 51.4% among patients in Saudi [186]. In yet another recent study, the virus was reported as 38% in Saudi patients [187]. All of these Saudi studies were conducted in Jeddah. 

From Makkah City, the reported epidemiological study identified 63.4% of DHF infection cases among Saudi nationals [188]. Similarly, a later study puts the estimate at more than 70% of Saudi nationals [189]. These previously published studies suggest that differences in proportions may exist between Saudi nationals infected with DHF virus in Jeddah and Makkah City. Contrary to previous data from Jeddah, in Makkah, it was clear that the majority of patients presenting with clinically significant DHF were Saudi nationals. Therefore, these results emphasized the fact that Saudi nationals are at greater risk of DHF infection. The awareness of these results is considered a cornerstone to enhancing the ability of healthcare professionals’ identification of the disease; and this might play an important role in the development of effective eradication strategies for the disease in Saudi Arabia localities.

Furthermore, the first cases of the virus, confirmed in Al-Madinah in 2008, showed that the isolated virus serotypes were DENV-1 and DENV-2 [190]. In 2009, the MOH in Saudi Arabia reported a total of 3350 cases of the DHF infection, with an estimated case fatality rate of about 4.6 per thousand in Saudi Arabia [171]. In August 2017, several countries in Asia, including Malaysia, Singapore, and Pakistan reported about 60,000, 1877, and 738 dengue cases including deaths, respectively. In the same period (2017), Saudi Arabia reported 39 confirmed dengue cases in Makkah, 19 of which occurred in August 2017, 60 suspected cases, and 15 cases pending laboratory confirmations. From these epidemic data indicating the reemergence of DHF infection in Saudi Arabia; Jeddah, Makkah, and Al-Madinah were shown to be the more susceptible areas, for this infectious disease, and this could be due to the fact that these cities are the sites of both the annual Hajj pilgrimage and/or the minor Umrah pilgrimage, which draw millions of Muslims to Saudi Arabia [171,190]. 

Currently, there are few epidemiological studies on DHF virus infection in Saudi Arabia. A study by Al-Azraqi et al. was conducted in 30 hospitals and 387 primary healthcare centers in two cities in the southern province of Saudi Arabia, particularly in Jizan, and Aseer. The study, which was limited to the seroprevalence among clinically suspected hospital-based patients, detected about 31.7% positive cases of dengue virus IgG among 965 randomly selected patients attending the outpatient clinics for any reason. The associated risk factors were male gender, younger age (15–29 years), lack of electricity, and having water basins in the house [191]. The authors suggested that the virus may occur in sporadic cases in Jizan, due to the nature of the city. Jizan is relatively flat and located at sea level [191]; thus the likelihood of the formation of small stagnant water following the rainfall in the city is high [171].

Interestingly, a retrospective cross-sectional study, which compared the clinical findings and/or the diagnostic laboratory results in uncomplicated patients, and patients who developed DHF, was conducted at Dr. Soliman Fakeeh Hospital in Jeddah, between January 2010 and June 2014. About 567 patients with a discharge diagnosis of DHF or dengue shock syndrome were identified [183]. Of these, 482 (85%) were adult patients within the age range 14–73 years, and 15% were children with age ranging from 2 months to 13 years. However, among all these patients, 67% of the adults and 63% of the pediatric cases were males. The clinical data from the hospital showed that in the adult patients, about 98% made a full recovery without complications while two patients died [183].

More recently in 28 January 2018, the MOH began an intensive campaign to eradicate the DHF virus from Saudi Arabian cities, to enhance public health awareness, and facilitate a change in hygiene behavior of citizens and residents. This resulted in a 50.7% reduction in the number of DHF infection among inpatient cases in Jeddah when compared to the same period in the previous year. However, the overall drop in DHF cases reached 38% in 2017, compared to the previous year [192]. Furthermore, recently, it is well-known that in Saudi Arabia, the DHF infection has been limited to the western and southwestern regions such as Jeddah and Makkah where *Aedes aegypti* exists. However, all DHF cases in Jeddah, Saudi Arabia, were mostly mild cases [171,175,192] and the prospect of dengue virus control lies with vector control, health education, and possibly vaccine use.

### 2.8. West Nile Fever

West Nile fever is one of the emerging zoonotic infections, which is caused by an arthropod-borne virus belonging to the genus *Flavivirus*, of the RNA family *Flaviviridae.* The virus’ main reservoir, which is responsible for the transmission of the disease, is the genus *Culex* mosquitoes [193,194]. The West Nile virus (WNV) derived the name from the site where the first case was isolated in 1937, from the blood of a woman with mild febrile illness living in the West Nile District of Uganda [195]. The first outbreak, in 1951–1952, was reported in Israel [196]. This constituted a turning point in the epidemiology of the virus, because it was thought to have originated from Israel following the introduction from Africa, and later introduced to the USA in 1999 [197,198]. Subsequently, the infection was documented across the globe [199], with the exception of Antarctica [194], in various species of vertebrates, including humans, mammals, non-human primates, birds, rodents, reptiles, and amphibians [200]. However, birds are considered as one of the main reservoirs of the virus [201]. Saudi Arabia is geographically close to several of the countries where WNV had circulated actively or had been reported; thus, there is a high risk of the disease being introduced into Saudi Arabia.

WNV is known to cause neurological disease in both humans and horses. However, the clinical manifestations of the disease in horses include ataxia, paralysis of the limbs, recumbency, hyperexcitability, and hyperesthesia. In Al-Ahsa, Saudi Arabia, a study was performed on 63 horses to test the incidence of the virus using the clinical examination and serologic ELISA test. However, from this previous study, while clinical examination for neurologic signs detected no significant findings, WNV antibodies were positively identified at serology among 33.3% of the tested population [202].

In 1999, Lanciotti et al. found this virus to be responsible for an outbreak of encephalitis in two fatal human cases from northeastern USA in late summer; and suggest a closely relation between this outbreak in the USA to a WNV infection which was isolated from a dead goose in Israel in 1998 [197]. The first cases of WNV in horses was identified in Egypt and France in the 1960s [203]; ever since, WNV has had significant public health impact worldwide due to its resurgence and dynamic epidemiologic features in humans and animals. Between 2008 and 2009, a study in Iran identified WNV antibodies in horses, and the results confirmed the highest activity of the virus reported in the Western and Southern Provinces with seroprevalences of up to 88% in some areas of Iran [204]. Although human cases and/or animal infections with WNV including horses have also been reported in Jordan and Lebanon (direct and close neighbors of Saudi Arabia) between 2000 and 2014 [205,206,207]; however, the reported WNV in patients or horses in these areas might have circulated in natural transmission cycles with close relationship to the WNV isolated from human and horses in Jordan, Lebanon, and Iran in 2000, 2010, and 2014, respectively. Humans and horses (incidental hosts), are unable to develop sufficient viremia to infect mosquitoes, hence, they are not included in the WNV lifecycle [203].

More recently, in 2016, using standard procedures, the Central Veterinary Research Laboratory in Dubai, the United Arab Emirates, described the first WNV isolation in a dromedary calf; and this supports the conclusion that WNV is present in the country [208]. The WNV zoonotic infection was probably transmitted through the human-animal interface; that is through the well-known contact with infected Arabian camels in Saudi Arabia. Interestingly, dromedary are exported from the United Arab Emirates to Saudi Arabia and vice versa; due to the closely related WNVs genes and their circulation through the natural transmission cycles worldwide, a complete genome sequencing for more WNVs strains, as well as comparative genomic and phylogenetic studies in Saudi Arabia, are needed to ascertain whether the dromedary infection with WNV exists in the country or not. However, the same facts have been suggested recently (2017), when it was suggested that WNV infection was introduced into Turkey at the time of the outbreaks in Saudi Arabia and Yemen. It was further suggested that the virus may have been introduced via unlawful entrance of viremic domestic or wild animals through the borders or through vectors that carry the virus into Turkey [209].

Camels play an important role in public health issues regarding zoonosis and they have been involved in most of the zoonotic infections which occurred in Saudi Arabia in the last three decades. They are reported as sources of infections—including rabies, MERS-CoV, Alkhurma virus, CCHFV, and RVF virus [52,94,101,108,120,124,210]—via direct physical contacts with camels and/or indirectly by having camels within or near the household in Saudi Arabia. However, some zoonotic infections among camels are sometimes asymptomatic; thus, they play a vital role in the mechanism of transmission of various diseases [211]. Furthermore, Wernery et al. reported that WNV can be transmitted by mosquito bites in different species including to humans, horses, camelids, and many other mammalian species as well as reptiles and birds [159,200,201]. To the best of our knowledge, there is still no extensive surveillance data regarding this disease among wildlife animals in Saudi Arabia. Strikingly, several of the human zoonotic cases that involve camels—which included different viral, bacterial, and parasitic infections on the Arabian Peninsula—have recently been highlighted as being caused by the consumption of unpasteurized camel milk [168].

## 3. Prevention and Control

Currently, in this review, some aspects of the most common viral diseases of zoonotic importance in Saudi Arabia were summarized; these are presented in Table 1. However, data regarding emerging and reemerging zoonotic viral diseases are reported as they occur from time to time from the same, new, and/or different localities from Saudi Arabia. While other viral zoonotic infections occur in other countries, which are considered to be close to Saudi Arabia, some infections spread to some localities within Saudi Arabia because of the geographical proximity as shown in Figure 1. 

Interestingly, some of these zoonotic viral pathogens were first exotic to Saudi Arabia (e.g., MERS-CoV and AHFV) and should be of more concern when reported in prevalence studies, and whenever they are detected by Saudi authorities. Epidemiological data should be focused more on both the trade routes and wildlife migration across the region, since these are potential risks for Saudi Arabia (e.g., from Yemen, Egypt, Gulf areas, and Sudan). Fortunately, there are many ways and/or approaches to improve the control of such different zoonotic pathogens in animals and humans in Saudi Arabia. However, the control measures of these viral zoonotic pathogens will not only benefit Saudi Arabia or Arabian Peninsula but will also be of high benefit to other countries, especially those with low prevalence, by stopping or controlling the spread of the epidemic worldwide. Prevention, control, and management of several zoonotic diseases usually require several important measures including the following. Having vaccination protocols for all suspected animal species by the use of up to date vaccines and compliance with the standards needed for all animals. Taking into account the highly needed and important investigation for these zoonotic viral diseases vectors, including vector breeding control (including vectors, hosts, and arthropods), and control of the animals (livestock) movements, with respect to trade and export [212,213]. Because an intensive livestock trade exists between Saudi Arabia and its neighboring countries, there may be increased risk of reemerging viral diseases of all kinds [214,215]. This is supported by several previous studies concerned with the route of livestock trade between Saudi Arabia and the neighboring countries (e.g., rabies through Yemen and/or Oman [36,216]; RVF through Kenya, Djibouti, and/or Egypt [127,149,212]; CCHF through Sudan [121]; influenza through Oman and Egypt [71,87,121,217,218,219]; WNV through Emirates, Egypt, Jordan, and Israel [196,197,199,203,208]; and DHFV through Egypt [190]; as well as MERS-CoV and AHFV viral infections, which originated and are transmitted globally from Saudi Arabia) [34,52,53,97,98]. 

Therefore, it is clear that a huge gap still exists in the sharing of published data about the acknowledged epidemiology of zoonotic diseases in Saudi Arabia, which rigorously prohibits speculations about the health burden of people. Currently, there are surveillance activities for some viral diseases—such as rabies, MERS-CoV, and influenza—but these are still being weakly addressed or neglected, especially at the human-animal interface. The important role of vaccination both in the prevention and control of animal diseases and the need to check the human sources in food or water must not be neglected. Also, management of animals, both outdoors and indoors must be taken seriously. However, owners of pets clinics and pets stores should be held responsible in ensuring that they keep their pets’ vaccination protocols up to date, and prevent any kind of animal behavior that might result in zoonotic risks to humans through bites or scratches by pets. Therefore, pet clinics and/or pets stores should be always considered a serious public health issue and vaccination should be obligatory. Therefore, the importance of the annual vaccination routine programs for all stray dogs against rabies, and regular investigation of other animals, should be considered.

In addition to this, pet clinics and stores should monitor pets’ health records, and their owners should be held fully responsible in ensuring that their animals remain healthy and fully vaccinated. This will guarantee for them and their neighbors a zoonotic disease-free environment (e.g., against rabies virus particularly in dogs). This is particularly important in view of the case of human rabies reported in March 2018 from a Makkah hospital. This involved a 60-year-old Saudi man who was admitted to the hospital with a history of an unprovoked scratch on his face by a dog. A month after his admission, his saliva PCR test confirmed rabies virus [41]. Nevertheless, rabies is endemic in animals in the Arabian Peninsula, with increasing numbers of reported cases form certain countries in the area including Saudi Arabia, Yemen, and Oman [36,41]. Kuwait, Qatar, and the United Arab Emirates are considered to be rabies-free, whereas there is no available information about Bahrain [216,220]. Furthermore, animal rabies cycle and cases reported in these endemic countries including Saudi Arabia are characterized by different animal species such as camels, cattle, goats, and sheep; however, the majority of cases are reported in feral dogs [36,216].

Fortunately, studies about pets with different zoonotic infections from pet clinics and/or pet stores in Saudi Arabia have been rarely detected among cats, dogs, and baboons. However, there was a previous study, which reported the occurrence of *Toxocara canis* infection in pets (dogs) in Riyadh, Saudi Arabia [30]. There were also two previous reports regarding a protozoan zoonotic infection of some pets with clinical manifestation, particularly in *Papio hamadryas* baboon in Riyadh [95,221]. In addition to this, another report highlighted the protozoan zoonotic infection in camels, in Riyadh [31]; however, more of these kind of studies are needed, because, they provide important opportunities to present a clear picture about indoor and outdoor animals and zoonotic pathogens such as viral, bacterial, fungal, etc. which involved, in Saudi Arabia.

By enhancing biosecurity and management in animal farms, the risk of reemerging pathogens particularly responsible for zoonotic diseases caused by viruses, can be reduced. This is a matter of economic importance; in view of the large livestock trade existing or that existed between countries in the Indian Ocean and Eastern Africa countries where several zoonotic diseases are endemic. However, a phylogenetic study strongly suggests that some zoonotic infections have been introduced into Saudi Arabia through ruminant trade [212,213]. Furthermore, following the adoption of the recommended guidelines of the World Organization for Animal Health through its Office International des Epizooties (OIE) Code, if such policies regarding the exportation and/or importation of animals are exactly followed, these would greatly limit the extent of this risk [214].

Furthermore, an emphasis should be made on surveillance to detect any sign of zoonotic disease that might occur in any animal kept directly in a quarantine station in any country of origin for 30 days prior to shipment to another country to ensure no clinical sign develops during that period. In addition, the longer quarantine periods or restriction of imported animals—particularly pets (e.g., dogs, cats, rodents, and monkeys) or goats, sheep, and camels—from endemic countries may be effective in reducing the introduction of zoonotic viruses. Of such measures, the control of vectors (e.g., ticks and mosquitoes), particularly the intermediate hosts and animal reservoirs, should be key components in the intervention strategy for zoonoses in Saudi Arabia. While the improving, enhancing, providing, and upgrading of laboratory techniques and/or testing in both veterinary and human medicines are fundamental to early detection and containing of any zoonotic disease or transmitted infection. 

Indeed, epidemiologic evidence should be linked with the seasonal time during the year for different zoonoses, and/or with any symptoms related to zoonotic infections that occur on the mainland a few years earlier. Up to date ecological factors on evolutionary issues, social movements, economic, and epidemiological mechanisms affecting zoonotic pathogens’ or their persistence and emergence, are not yet well understood. However, studies on the ecological, socioeconomic, and health issues are needed to assess the sustainability and acceptability of measures by breeders, as well as information that ensures appropriate slaughtering or consumption practices, which will decrease the risk of infection to humans [200]. Due to these facts about the ecological cascade and evolutionary perspectives, authorities can provide valuable insights into pathogen ecology and can inform zoonotic disease control programs; and thus evaluate their global effect in terms of actual disease and its socioeconomic correlations.

Enhancing biosecurity and management in the treatments of various zoonotic infections may result in appropriate use of vaccinations, drugs, and antibiotics, however, the overuse of these agents result in various types of resistance. Furthermore, regardless of the influenza virus resistance level to treatment, according to a serosurveillance, the enzootic influenza virus H5N1 in Egypt is endemic [87]. The same result to oseltamivir-resistant influenza viruses are reported globally, with a high susceptibility to these antiviral drugs among all reported cases of the virus from Egypt. Resistance was also found in most infected viral cases that are usually acquired in humans through intensive contacts, particularly with backyard birds, among women and children [82,83,86,87]. Therefore, drug regimens in Saudi must include vaccines against this virus during Hajj and Umrah seasons, for Egyptians.

Most importantly, epidemic zoonotic cases of influenza among pets has highlighted the importance of circulating influenza viruses globally, and the importance of ensuring the effective use of antivirals for the prophylaxis and treatment of influenza, especially because of the increased number of new pets stores in Saudi Arabia, particularly in Riyadh [74,86]. Thus, studies on drug resistance are considered to be of a high public health importance, although, this might demonstrate the best scenario of how drug resistance in Saudi Arabia can pave its own way and/or role into the reemerging of different zoonotic pathogens.

On the other hand, few studies have been done in this area to identify the relationship between different gatherings and the occurrence of signs and clinical symptoms of viral infections, especially among humans of different ages and gender. However, there are several suggestions and information regarding zoonoses (e.g., influenza and MERS-CoV infections) in Saudi Arabia among the elderly, based on age and gender [63,222]. More recently, increased availability of limited public health data on the prevalence of some zoonotic diseases and associated risk factors or data that identifies the relationship between different zoonotic pathogen antibodies in pregnant women, are of importance [223,224].

Central to the profound worldwide changes in religious beliefs and activities is the birth of a new era of both emerging and reemerging diseases that could be arranged under the umbrella of social movements, along with its own role in the spread of zoonotic diseases. Thus, any prevention and/or control strategies against any zoonotic pathogen have to take this point of view into account. Furthermore, annually, Saudi Arabia hosts the largest international gathering of Hajj where many millions gather in a small geographical area. This puts Saudi Arabia in the front line of threats of pandemic diseases [215]. Thus, Saudi Arabia must keep a high level of alertness in monitoring the situation of these pathogens, particularly in view of the potential for global spread of pandemic viruses especially during winter and around the Hajj season (e.g., MERS-CoV infections, AHFV, and influenza viruses). Therefore, there is need to prevent further spread of the virus locally, regionally, and internationally. Interestingly, with WNV outbreaks, the Israeli-like WNV that was isolated in white storks in Egypt in 1997–2000 suggests that migrating birds do play a crucial role in the geographical spread of the virus [225]. Recently, the same fact was again suggested in 2017, when the same infection by this virus was introduced into Turkey at the time of the outbreaks in Saudi Arabia and Yemen; it was stated that the WNV virus might have been introduced via unlawful entry of the viremic domestic or wild animals through the borders, or by vectors carrying the virus to Turkey [209].

More recently, epidemiological data of zoonotic viral pathogens from Saudi Arabia and/or from other neighboring countries after it was confirmed through laboratory test isolation from dromedaries (e.g., rabies, MERS-CoV, RVFV, and WNV) may enhance a high interest in the search for other novel zoonotic viruses in dromedaries [36,52,208,211,226,227]. Furthermore, the habits of ingestion off unpasteurized milk from camels as a rare delicacy by Saudi people need to be checked. Moreover, viral pathogens such as RVFV are acquired through the importation of camels, while the remaining pathogens (e.g., rabies and influenza viruses) are endemic worldwide. Of these (e.g., influenza virus), there is need for a highly preventive zoonotic control in Saudi, due to fact that the isolation and genetic characterization of H5N1 was reported in 2017 among vaccinated meat-turkeys flock in Egypt, a neighboring country, that was previously reported to have more than 100,000 travelers to Saudi Arabia during Hajj pilgrimage seasons, annually [219]. This might be considered as one of such important risk factor of possible introduction or spread of influenza pathogen in Saudi Arabia [218,219].

Lastly, increased zoonotic pathogens surveillance, particularly influenza, during the Hajj season, increased infection control interventions, screening, and quarantine of suspected cases, provision of adequate medical treatment, sustainable awareness, increased education and training of target groups at high risk (e.g., doctors, nurses, veterinarians, and animal workers such as farmers and abattoir workers, etc.) are of great importance to reduce the burden of zoonoses among Saudi Arabian localities. Fortunately, in collaboration with three organizations—including the MOH in Saudi Arabia, the USA Centers for Disease Control and prevention, and the WHO—a successful preparedness plan during the Hajj season was put in place to vaccinate all pilgrims before leaving their home countries [68]. Altogether, there is an urgent need for collaborative surveillance and intervention plans for the control of zoonotic pathogens in Saudi Arabia.

## 4. Summary and Conclusions

With Saudi Arabia, the focal point of the ongoing zoonotic pathogens outbreak could be due to the large number of religious pilgrims congregating annually particularly in Makkah, Jeddah, and Al-Madinah, the main three cities for Hajj and Umrah, which drastically increased the potential for uncontrolled global spread of zoonotic infections [168].A zoonotic pathogen outbreak could be dramatically decreased among the annual Saudi pilgrims if we take into account the fact that: Jeddah Governorate, the main seaport in Saudi Arabia is considered to be the main entry point for over 2 million pilgrims coming for Hajj or Umrah annually. All these numbers of pilgrims arrive through the Jeddah Islamic Port before going on to Makkah, for the start of their Umrah and/or Hajj. Surprisingly, the current review showed that during an outbreak, each of these eight most zoonotic viruses (rabies, MERS-CoV, influenza, AHFV, CCHFV, RVFV, DHFV, and WNV) which occurred and/or cases confirmed in Saudi Arabia particularly from (Jeddah and/or Makkah) areas with at least one or all of these eight zoonotic viral pathogenic diseases [33,44,46,78,96,97,98,99,121,130,156,171]. The spread could also have been due to the fact that Jeddah is the main port for animal importation to Saudi Arabia. At the same time, it is the closest area to several countries where some zoonotic outbreaks were reported. To enhance this spread, the role of the active circulation of zoonotic viruses, during their natural transmission cycle, has been reported, however, an importation might increase risk of disease introduction to Saudi Arabia.Almost annually, from the more than 7 million pilgrims who come to Makkah and Madinah from different countries worldwide during Hajj and Umrah, the Kingdom’s revenue in 2012 was put at more than 62 billion Saudi Riyals (~ about 16.5 billion US Dollars), 10% up from the 2011 figures. This Hajj revenue accounted for 3% of the gross domestic product for the Kingdom of Saudi Arabia. To avert all that number of health hazards from zoonotic diseases in view of economic facts, the global community and particularly the pilgrims need more gift items made in Saudi Arabia to control and prevent the spread of zoonotic diseases which could be transmitted among Hajj and Umrah pilgrims.

Therefore, the following recommendations are suggested in order to improve public awareness and/or health education of zoonotic viral diseases in Saudi Arabia: 

Based on findings of previous studies, health education strategies could enhance the awareness of the Saudi population regarding viral zoonotic diseases through health education program experiences of other countries, particularly during Hajj and Umrah seasons. This response can draw on the availability of several studies on how to improve, control, and prevent the spread of several zoonoses in both animals and humans, worldwide [78,96,97,98,99].

Public health authorities must highlight the importance of promoting health education and facilitate the outcomes of studies for reducing patient cases in Saudi Arabia. The Saudi authorities and government bodies such as the MOH should also launch different programs and workshops to increase public awareness about these zoonotic infections. This should involve the cooperation of the Saudi regime, and the private and public sectors. Different activities may be needed in Saudi Arabia—such as the practice of self-protection against these diseases, adult control strategies, control activities, and regular workshops—to achieve control and prevention.

Enhancing of self-awareness among people through health education programs or other strategies for the prevention of viral zoonotic diseases, which require vectors (such as mosquitoes, ticks, and fleas) for their transmission; are important issues on which the Saudi population should be educated. They should also be educated about the adverse effects of arbitrary application of insecticides without prior knowledge on dose, resistance, and side effects. Increasing the knowledge about the biology and ecology of the animal vectors in society is also crucial. Furthermore, the Saudi Ministry of Culture and Information should establish intensive health education programs on television channels, radio, and newspapers to increase public awareness and to maintain hygiene conditions within the Kingdom and in Saudi houses. The Saudi Ministry of Agriculture could play a big role by regularly controlling the application of vaccinations and/or antibiotics on animals which used in the veterinary sector, and also accounting the misuse of such agents following other developed and developing countries on controlling and/or accounting drug strategies [87,228,229]. Thus, veterinary regulations of animal antibiotics—including overuse of drugs and their application—must be enforced to alleviate the serious public health problems.

## Figures and Tables

**Figure 1 pathogens-08-00025-f001:**
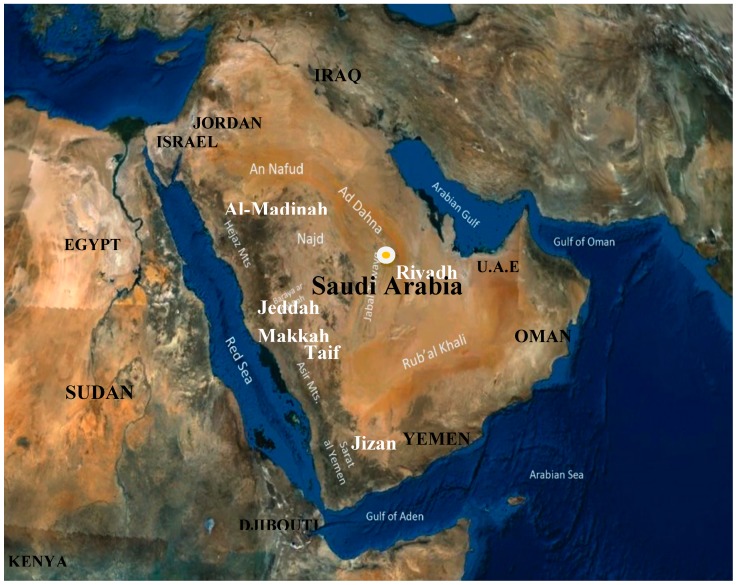
Map of Saudi Arabia indicating the most reported areas with viral zoonotic infectious cases in different regions within Saudi Arabia and abroad (comparing to between countries with common borders with Saudi), to show where zoonotic viruses were first isolated.

**Table 1 pathogens-08-00025-t001:** Summary of the zoonotic infectious viral diseases reported in Saudi Arabia to be causing different outbreaks, by affected area and year

Pathogens *	First Country Affected and Year	First Cases Recorded from Saudi Arabia, Year and Area	Reference on Cases Information from Saudi	Pathogen Sources ^+^	Pathogen Host
Rabies	First human rabies death in the USA, 1899	2018 in Makkah	[41]	Saliva, CNS	All mammals
MERS-CoV	Saudi Arabia in 2012	2012 in Jeddah, Riyadh, and Makkah	[34,55,61,63]	Nasal and eyes discharges	Arabian camel
Influenza A H1N1; H5N1	First in 1918, then 1976 in the USA. Another fatality complication occurred from 2005–2009	2009 in Riyadh, Eastern region, and Jeddah	[68,78,84,87]	Aerosols, bird feces	Birds + pigs + horses + dogs + sea mammals + humans
AHFV	Saudi Arabia in 1994	1994 in Jeddah, and between 2006–2009 in Jeddah, Makkah, and Najran	[33,90,95,96]	Blood, feces, and nasal discharges	Sheep + Arabian camel
CCHV	Crimean Peninsula in 1944 and 1956 from a Congo child	1989–1990 in Makkah. 1991–1993 in Makkah, Jeddah, and Taif. 2009 in Jazan	[114,115,116,117]	Blood, tissue, and ticks	Domestic ruminants
RVFV	Great Rift Valley of Kenya in 1912	2000 in Southwestern regions of Saudi (Jizan and Aseer)	[148]	Blood, tissues, and mosquitoes	Lamb + goat + bovine + Arabian camel
DHFV	Before the 18th century. Manila in 1953. Jakarta Indonesia and Egypt in 1779	1994 in Jeddah, 2004 in Makkah, and 2008 in Al-Madinah	[171,177,178,179,180,181,182,184,205,209]	Blood, tissue, and mosquitoes	Human
WNV	Uganda in 1937	2007 in Al-Ahsa, virus detection was by serology positivity	[196]	CNS, tissue	Horse + human

Pathogens *: MERS-CoV = Middle East respiratory syndrome coronavirus. AHFV = Alkhurma hemorrhagic fever virus. CCHV = Crimean-Congo hemorrhagic fever virus. RVFV = Rift Valley ever virus. DHFV = Dengue hemorrhagic fever virus. WNV = West Nile virus. Pathogen source **^+^** of rabies virus, influenza virus, AHFV, RVF virus, and WNV were reported from a previous study on several zoonotic diseases diagnosed in the Arabian Peninsula [23]. Saudi Arabia ^#^ was reported as the first country affected by these two viruses (MERS-CoV and AHFV). CNS = central nervous system.

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
