# Peer review of "An Overview of the Most Significant Zoonotic Viral Pathogens Transmitted from Animal to Human in Saudi Arabia"

_pathogens, 2019, doi:10.3390/pathogens8010025_

Round 1

Reviewer 1 Report

Dear Authors

The purpose of a review paper is to succinctly review recent progress in a particular topic. Overall, the paper summarizes the current state of knowledge of most important viral zoonoses pathogens in Saudi Arabia. It creates an understanding of the topic for the reader by discussing the findings presented in recent research papers. The topic addressed in the manuscript treats an important concern in public health. Old and recent articles are referred. The study from a scientific point of view seems to be well done.

Figure 1 – The legend of Israel and Jordan is not visible, please correct it.

Author Response

Response to Reviewer 1 Comments and Suggestions 

Point 1: The study from a scientific point of view seems to be well done. However, in Figure 1. The legend of Israel and Jordan is not visible, please correct it.

Response 1: THANKS very much ended for that supportive comments. Regarding the legend of these two countries in Figure 1- had been checked and corrected to be easyely visible. 

Reviewer 2 Report

The manuscript is very hard to follow due to the large number of grammatical errors. Several statements, throughout the text, seem misleading and/or are incorrect. The figure and tables don't really add a lot to the manuscript. The manuscript could be improved by focusing more specifically on the situation in Saudi Arabia and making the Table reflect this vs. making comments about where a particular disease was first reported. General comments about influenza, West Nile virus and other zoonoses don't really add a lot to the manuscript.

Author Response

Response to Reviewer 1 Comments

Point 1: The manuscript is very hard to follow due to the large number of grammatical errors. Several statements, throughout the text, seem misleading and/or are incorrect. The figure and tables don't really add a lot to the manuscript. The manuscript could be improved by focusing more specifically on the situation in Saudi Arabia and making the Table reflect this vs. making comments about where a particular disease was first reported. General comments about influenza, West Nile virus and other zoonoses don't really add a lot to the manuscript.

Response 1:

It seems we were far away from writing a good text just because I myself did all the manuscript "is very hard to follow, misleading, figures and tables as well don't really add a lot to the manuscript" all that when I read it made me in very hectic times. However, it seems the reply to the first referee can be easily stated as shown. Our full reply should go directly to the editor stating that:

 Reply to the comments of the (Reviewer 2):

It considered to me as a long one and needs to require and/or modification all the manuscript that:

Several statements, throughout the text, seem misleading and/or are incorrect might be due to the fact that I'm from overseas countries and not well establishing my work but I think the manuscript is very important to the public health issue and could pave the way to more effective health policy needed in this area. We would greatly appreciate if the text published regarding its scientific data contains, however, we would greatly appreciate if any exits and if any reference, to any previous work in this specific area could be pointed out to us to look for as a model of how to write such a manuscript and we look through for sure and we are almost not so bad on that issue.

The figure and table reflect the importance of zoonoses control policy in and out Saudi Arabia in the same time, however, two of these viral occur for the first time in Saudi Arabia and then distributed to the globe. In additions, the other 6 viruses almost from the countries around Saudi Arabia as shown in our figure and table (putting into accounts there are many other viral zoonoses but these 8 diseases are the most occurrence in this area) and this is exactly the situation in Saudi Arabia which almost could be controlled easily if in and out zoonotic focused regularly particularly with the neighbors country; and that’s why we mention this through figure and table as shown in the text.

-          Comments about influenza, West Nile virus and other zoonoses add a lot to the manuscript because of influenza particularly prevalence as first

1.       Influenza viruses H5N1 in Egypt is an endemic viral disease. Also, outcomes of swine influenza virus from Egypt was laboratory diagnostic and confirmed very early (1979-1980). Moreover, high numbers of Egyptian Muslims were continuously coming during the year for doing Umrah and/or once for Hajj pilgrimage season in Saudi Arabia; however, the poultry industry there was estimated in 2006 to be one billion birds with several millions of labors. This is considered highly important to Saudi Arabia. Interestingly, surveying data regarding this virus showed ahigh number of Egyptian Muslims during once for Hajj pilgrimage season with highly susceptible to antiviral drugs and thus it could play a major role in the introduction of new influenza viruses not only to Saudi Arabia but also to other countries on this planet. Unfortunately, there is no such program in Saudi Arabia, which could pose several public health concerns.

2.       WNV outbreaks occur in Israeli and then isolated in Egypt in 1997–2000 suggests that migrating birds do play a crucial role in the geographical spread of the virus, while WNV geographically was well-known to emerge through Emirates, Egypt, Jordan, and Israel to Saudi Arabia.

-          Reply to a large number of grammatical errors:

As instructed by the referee the manuscript has been reviewed in its entirety for the appropriate use of the English language, grammar scientific terms, and acronyms. We want to confirm that the changes needed were not many and the scientific content was kept intact when the linguistic changes were made.

My best regard,

Round 2

Reviewer 2 Report

There is some interesting information presented but I would suggest that the title be changed and that the rest of the paper be edited for grammatical errors.

Please see the attached draft with some suggested edits

Author Response

Response to Reviewer 2; Round 2 â€¨ 

Response 1: 

A point-by-point response to the reviewer’s 2 comments.

Regarding this Review [Pathogens-403280],

Dear Reviewer 2;

Hereby as possible I'm almost checked all that important points you mentioned in your extensive review [Reviewer 2- Round 2], however, it takes time and I had to read every single word in my review again and again then start making all the changes which required from your end; starting from the Title to References through adding a lot of the important ideas to the review text and checked all the grammar errors as well.

Thanks very much ended because you make it easier for me to find out where exactly I wanna make some editing, corrections, changing and/or adding a new data or more such details in this review. However, at the end of each point there, you make the real value to this work come true and thus this review according to your requests I think right now might be almost able to show its highly valuable outcomes and can also pave the way to offer a significant impacts according to new points and idea which finally were added too cause of your valuable comments.

Kind Regards

Dr. Omar Al-Tayib

Round 3

Reviewer 2 Report

The paper has been improved but there are still numerous grammatical errors, can you find help from a professional proof reader or editor ?

Author Response

I thoughts all grammatical errors had been checked from my end and corrected in Round 2 text which send to you last week according to the Review 2 Round 2. Regards